# Space-confined synthesis of sinter-resistant high-entropy nanoparticle library

Shaoqing Chen [1,2], Xia Li [2] ✉, Ziqiang Qu[2], Xiang Li [3], Yuanzhu Gao[4], Peng-Fei Liu [5,6], Zhi-Qiang Dong [3], Peng Yu [3], Qiming Sun [2] ✉, Shixue Dou[7], Zhongfan Liu [8] & Jingyu Sun [1] ✉

The tailorable confinement of high-entropy nanoparticles (HE-NPs) within molecular sieves (HE-NPs@MSs), synergizing merits of cocktail effects and geometric polymorphs, holds potential for advancing heterogeneous catalysis. However, effective and universal synthesis affording size homogeneity and production scalability remains elusive. In this contribution, we present a versatile strategy for encapsulating ultrafine HE-NPs within diverse mesoporous/microporous MSs to enable the rational construction of HE-NPs@MS library. By utilizing the approach of quenching space-confined liquid metal droplets, the resulting HE-NPs@MSs comprise anti-sintered HE-NPs (1 to 5 nm in diameter) with narrow size distributions. As a proof-of-concept demonstration, a HE-NPs@MS prototype catalyst containing trace amounts of Pt is employed in the propane dehydrogenation reaction, achieving a propylene formation rate of up to 44.2 mol $g_{Pt}^{-1}$ $h^{-1}$, which is 31.6 times greater than that of the monometallic Pt@MS counterpart. Our strategy facilitates high-throughput synthesis and large-scale production, opening tantalizing opportunities in the utilization of high-entropy nanomaterials for various applications.

High-entropy materials (HEMs), composed of five or more principal elements (each with atomic percentages ranging from 5% to 35%), exhibit a configurational entropy surpassing a specific threshold (1.5 R, where R represents the ideal gas constant). These materials hold great promise for various applications, including heterogeneous catalysis[1–4]. The key characteristics of HEMs, such as the high-entropy impact, local lattice distortions, sluggish diffusion, and cocktail effects, provide a versatile design toolkit for catalysts. Notably, HEMs showcase the potential to replace scarce elements with more abundant congeners[4,5]. Ultrafine high-entropy nanoparticles (HE-NPs) with size uniformity are expected to offer a diverse range of adsorption and reaction platforms by enabling access to polynary surface/subsurface atom arrangements[6–8]. In this context, the adsorption strengths of reaction intermediates can, in principle, be continuously tuned to identify suitable active sites[5,9]. While the high-entropy impact helps stabilize nanoparticles, ultrafine HE-NPs remain susceptible to sintering at elevated temperatures[10].

Molecular sieves (MSs) not only provide a confined space to anchor metal species in preventing aggregation but also offer shape selectivity as reliable host materials. Metal species embedded within

[1]College of Energy, Soochow Institute for Energy and Materials Innovations, Key Laboratory of Advanced Carbon Materials and Wearable Energy Technologies of Jiangsu Province, Soochow University, Suzhou, PR China. [2]Innovation Center for Chemical Science, College of Chemistry, Chemical Engineering and Materials Science, Jiangsu Key Laboratory of Advanced Negative Carbon Technologies, Soochow University, Suzhou, PR China. [3]Department of Mechanics and Aerospace Engineering, Southern University of Science and Technology, Shenzhen, PR China. [4]Cryo-electron Microscopy Center, Southern University of Science and Technology, Shenzhen, Guangdong, PR China. [5]Institute of High Energy Physics, Chinese Academy of Sciences, Beijing, PR China. [6]Spallation Neutron Source Science Center, Dongguan, PR China. [7]Institute of Energy Materials Science, University of Shanghai for Science and Technology, Shanghai, PR China. [8]Center for Nanochemistry, College of Chemistry and Molecular Engineering, Peking University, Beijing, PR China. ✉e-mail: xiali@suda.edu.cn; sunqiming@suda.edu.cn; sunjy86@suda.edu.cn

MSs featuring well-ordered nanoporous architectures–denoted as Metal@MSs–are of growing interest as heterogeneous catalysts in petroleum refining and petrochemical industries. Moreover, ultrafine HE-NPs@MS materials are anticipated to exhibit superior properties due to the synergistic effects of the active metal regimes and versatile MS supports[11–15]. Recently, highly dispersed HE-NPs were synthesized via wet-chemical methods using surfactants[16–21]. Surfactant-stabilized HE-NPs face challenges in infiltrating the nanopores of inorganic molecular sieves via conventional impregnation methods. In situ preparation, where pre-synthesized high-entropy alloys are introduced into the mother liquor during the synthesis of the molecular sieve, often results in structural damage to the high-entropy alloys and significant phase separation[11,12]. Another approach involves thermal shock, which entails rapid heating to ~900 °C followed by quenching[22,23]. Metals prepared via these methods are typically located on the outer surface and exhibit uneven size distribution. In response to previous endeavors where HE-NPs were loaded onto the support surface (Fig. 1a), there is an urgent need to develop emerging strategies to confine ultrafine HE-NPs within the well-ordered nanopores of MSs toward improved dispersion and stability (Fig. 1b). Such a route should be easily implementable, versatile enough to facilitate the development of a product library, supportive of high-throughput screening, and scalable for large-scale manufacturing[11].

In this work, we present a generic and scalable synthesis strategy based on the quenching of space-confined liquid metal droplets to construct a diverse library of sinter-resistant HE-NPs@MSs, featuring ultrafine high-entropy nanoparticles confined within molecular sieves. Guided by theoretical predictions of droplet growth behavior on open surface or in confined space, an ICQ approach, comprising incipient wetness impregnation (I), short-time calcination (C), and rapid quenching (Q), is developed. Such a method enables to yield uniformly distributed HE-NPs with diameters of 1–5 nm and narrow size distributions, which is applicable across a wide range of metal compositions and molecular sieve frameworks. To validate the effectiveness of the proposed strategy, a Pt-containing HE-NPs@MS catalyst is employed in the propane dehydrogenation reaction (PDH), achieving a propylene formation rate of up to 44.2 mol gPt$^{-1}$ h$^{-1}$, which is 31.6 times higher than that of its monometallic Pt@MS counterpart. Upon PDH reaction test at 550 °C, the HE-NPs remain a confined status in the molecular sieve matrix, with Pt species preserved in atomically dispersed form, underscoring the excellent anti-sintering property of the catalyst.

## Results

### Design principle of HE-NPs@MSs

The determining factors for the size of HE-NPs are first analyzed in comparative cases of opened- and confined-space. Considering that multi-metal salt precursors could rapidly decompose within seconds to generate liquid metals at 900 °C, HE-NPs were known to be formed by "freezing" the liquid metal nanodroplets via a quenching process[22,24–26]. The nanodroplets tend to aggregate into larger sizes over a non-wetting surface to reduce the total surface energy. The surface Gibbs free energy ($G_R$) of a droplet with radius $R$ is (Eq. 1):

$$G_R = \gamma_{LV}A_{LV,R} + \gamma_{SL}A_{SL,R} - \gamma_{SV}A_{SV,R} \qquad (1)$$

where $\gamma_{LV}$, $\gamma_{SL}$, $\gamma_{SV}$ and $A_{LV,R}$, $A_{SL,R}$, $A_{SV,R}$ is the interfacial tension and area of liquid-vapor (LV), solid-liquid (SL), solid-vapor (SV) interface, respectively.

According to Young's equation, the relationship between interfacial tension is (Eq. 2):

$$\gamma_{SV} = \gamma_{SL} + \gamma_{LV} \cos\theta \qquad (2)$$

where $\theta$ is the contact angle of the solid-liquid interface ($\theta > 90°$ on a non-wetting surface, Supplementary Fig. 1).

If the surface is an open non-wetting space, the total surface energy ($G_{Total}$) is inversely proportional to the droplet radius (Eq. 3 and

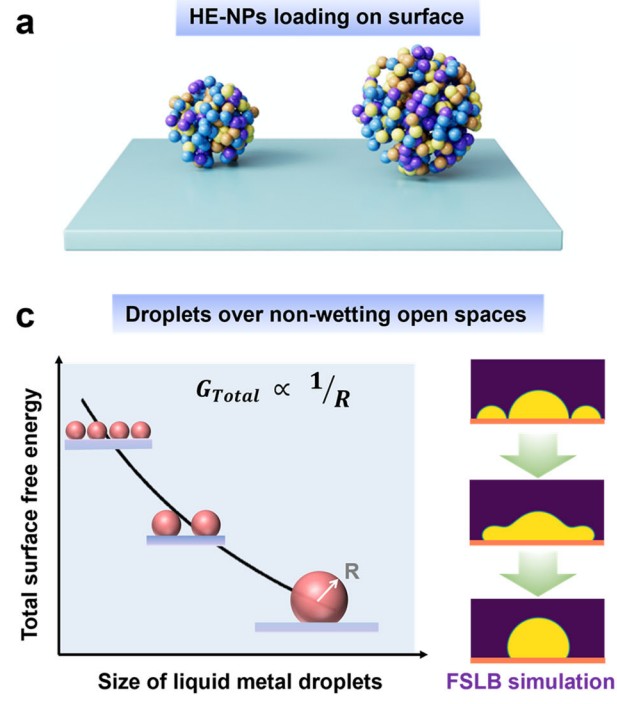

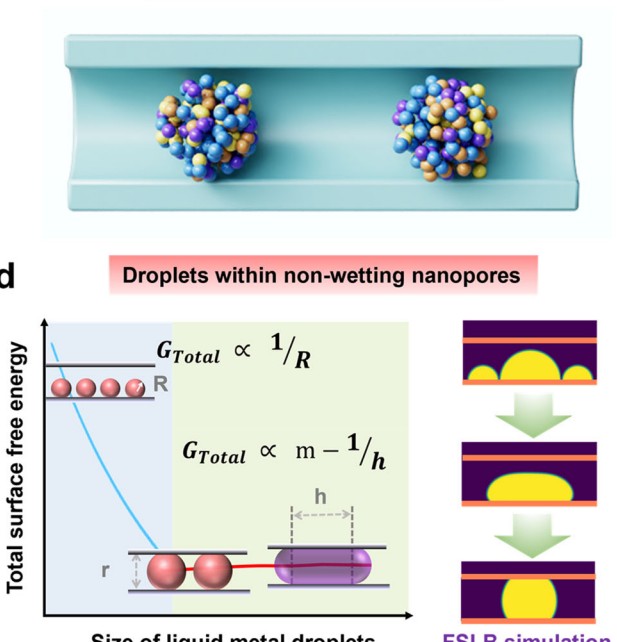

**Fig. 1 | Theoretical insight into the formation.** Schematic diagram of high-entropy nanoparticles **a** loading on the surface and **b** confining within the nanopores. The functional relationship between the size of liquid metal droplets on non-wetting surfaces and total surface Gibbs free energy, with corresponding simulation of the interface behavior of liquid metal droplet by FSLB method **c** on an open surface; **d** within a nanopore.

Supplementary Eq 1–11), causing the droplet to spontaneously grow (Fig. 1c and Supplementary Movie 1). Our computational simulations employing fractional step lattice Boltzmann (FSLB) method confirm the thermodynamically driven spontaneous growth process (Fig. 1c). It is worth noting that the FSLB approach used here primarily captures qualitative trends in morphological evolution on open surfaces and within nanopores. As for quantitative insights into the behavior of heterogeneous nanoalloys, i.e., the affinity between specific elements and the substrate, complementary methods, including molecular dynamics simulations, would be necessary. The final size of a droplet is limited by diffusion kinetics and is related to the growth time and precursor concentration. And the irregular distribution of defects on the substrate surface might lead to uneven droplet sizes, ultimately resulting in non-uniform nanoparticle size distributions of the final product[22].

$$G_{Total} \propto \frac{1}{R} \qquad (3)$$

As for a confined non-wetting space, when the droplets are smaller than the pore size, the growth behavior is analogous to that in an opened surface, and the droplets would spontaneously grow to a size equivalent to the pore size (Fig. 1d and Supplementary Movie 2). When the size of the droplet is larger than the pore size, the droplet would form a geometric combination with two hemispheres at both ends and a cylinder at the middle during growth because of the confinement effect, accompanied by a slight increase in $G_{Total}$. This is in stark contrast to the thermodynamic trend for opened surfaces. The FSLB simulation reveals the process of spontaneous growth of small droplets into a size equivalent to the pore size, as depicted in Fig. 1d. Then, the $G_{Total}$ is positively correlated with a constant minus the reciprocal of the length ($h$) of the middle cylinder.

$$G_{Total} \propto \left( m - \frac{1}{h} \right) \qquad (4)$$

where $m$ is a constant related to surface tension of a liquid metal (Eq. 4 and Supplementary Eqs 12–16).

In terms of multiple droplets having the same size equivalent to the pore, the FSLB simulation further suggests that they tend to remain independently (Supplementary Fig. 2 and Supplementary Movie 3). It could be inferred that the key to preparing HE-NPs@MSs lies in allowing liquid metal to exist within the pores and maintain a random elemental dispersion. Note that liquid metals with high surface energy would be difficult to enter non-wetting nanopores of MSs (alike water droplet driven by capillary forces[27,28], the *on-site* formation of liquid metal droplets within the nanopores is highly desirable.

Guided by the above discussions, our MS-confined, in situ synthetic protocol involves an incipient wetness impregnation (I) of multi-metal salt solutions into the MSs pores, a short-time (~60 s) calcination (C) at 900 °C, and a rapid quench (Q) under ice-water, as illustrated in Fig. 2a. Employing the ICQ strategy, high-entropy nanoparticles affording up to eleven types of metallic elements could be produced within MSs. The ultrafine nanoparticles (1 to 5 nm in diameter) harvest a narrow size distribution (± 20%), which are uniformly dispersed in ordered pores of mesoporous or microporous MSs possessing distinct framework architectures. Such an ICQ synthesis could be readily scaled up by producing over 20 g HE-NPs@MSs materials within 5 min (Fig. 2b and Supplementary Fig. 3).

As for a prototype synthesis of high-entropy oxide (HEO), Pt-Quinary-HEOs@MCM-41, a mixed solution containing various metal chlorides (Mn, Fe, Co, Cu, and In in equal molar ratios along with $H_2PtCl_6$ at a 1/12.5 molar ratio) was impregnated into the mesopores of MCM-41 (a mesoporous MS). The metal salts@MCM-41 precursor was rapidly heated to 900 °C and quenched in ice-water. Scanning

transmission electron microscopy (STEM) and energy dispersive X-ray spectroscopy (EDS) inspections demonstrate that the prepared nanoparticles with a narrow size distribution ($3.0 \pm 0.5$ nm) are homogeneously anchored within the nanopores of MCM-41 (Fig. 2c and Supplementary Figs. 4–9). The representative high-magnification STEM image in Fig. 2d with electron tomography imply that the high-entropy nanoparticles are confined into the pores of mesoporous MCM-41 (Supplementary Fig. 5 and Supplementary Movie 4). The average particle size matches the pore dimension of MCM-41, consisting with our theoretical analysis (Supplementary Fig. 6). The elemental mappings and line scan profiles depict the homogeneous distribution of multiple metal species within individual nanoparticles of Pt-Quinary-HEOs@MCM-41 (Supplementary Figs. 7 and 8). The molar ratio of each non-noble metal element within the composite is ~1:1 (Supplementary Fig. 9). $N_2$ adsorption/desorption analysis further suggests that the MCM-41 pore structures remain intact during the ICQ process (Supplementary Fig. 10 and Supplementary Table 1). In parallel, microporous zeolites (ZSM-5 with the MFI zeotype) were also examined as the host. The STEM images and electron tomography show the morphology of synthesized Pt-Senary-HEOs@ZSM-5 (Fig. 2e; Supplementary Figs. 11 and 12 and Supplementary Movie 5), where uniform nanoparticles with an average size of $4.0 \pm 0.8$ nm are anchored inside ZSM-5 crystal (Supplementary Fig. 13). No metal particles are observed on the edge of 2 d projections of ZSM-5 particles from STEM, and the elements Pt, Mn, Fe, Co, Cu, Zn, and In were found to be homogeneously distributed throughout the material (Supplementary Fig. 14). Figure 2f and Supplementary Fig. 11 display several irregular grooves resembling earthworm holes, which could be attributed to the flow erosion of liquid metal. This erosion partially disrupts the MS framework, creating breakages that extend across several micropores to form mesopores and defects (Supplementary Fig. 15). The "earthworm holes" create additional spaces that accommodate nanoparticles larger than the pore size of ZSM-5, while the overwhelming majority of the micropores remain intact, displaying a well-defined lattice pattern (Supplementary Figs. 11–17 and Supplementary Table 2).

The high-entropy nanoparticles synthesized within MSs using the ICQ strategy exhibit remarkably uniform sizes, ranging from 1 to 5 nm with a narrow size distribution (±20%). In this context, a comparison of nanoparticle sizes is made between this study and recent literatures that employed wet chemical synthesis, calcination method, and thermal shock method (Fig. 2g and Supplementary Table 3)[9,10,16–26,29–39]. The size uniformity achieved in this work is clearly superior to that of HE-NPs prepared via high-temperature strategies and is comparable to those produced by wet-chemical methods, which require precise control. While ultrafine HE-NPs could also be attained by wet-chemical treatments, they tend to agglomerate during heat treatment, limiting their stability and practical applications. The ICQ method is otherwise carried out at 900 °C, ensuring the structural stability of HE-NPs@MSs under elevated temperatures. This approach allows for the general and scalable production of HE-NPs, providing excellent spatial dispersion and size uniformity. Additionally, the relatively mild synthesis conditions and low manufacturing costs make the ICQ method highly suitable for scalable production.

## Synthetic essence of HE-NPs@MSs

During the ICQ process, the impregnation stage is critical in determining whether the metal species can infiltrate the ordered pores and form liquid droplets within them. The cooling stage is essential in facilitating the formation of the HE-NPs without phase separation. To highlight the superiority of the ICQ strategy in the tailorable synthesis of HE-NPs@MSs, we closely examine the detailed morphology of particulates grown from comparative recipes. As for traditional impregnation method, the multi-metal salts are prone to be positioned over the outer surface of the MSs substrates[13]. As shown in Fig. 3a and

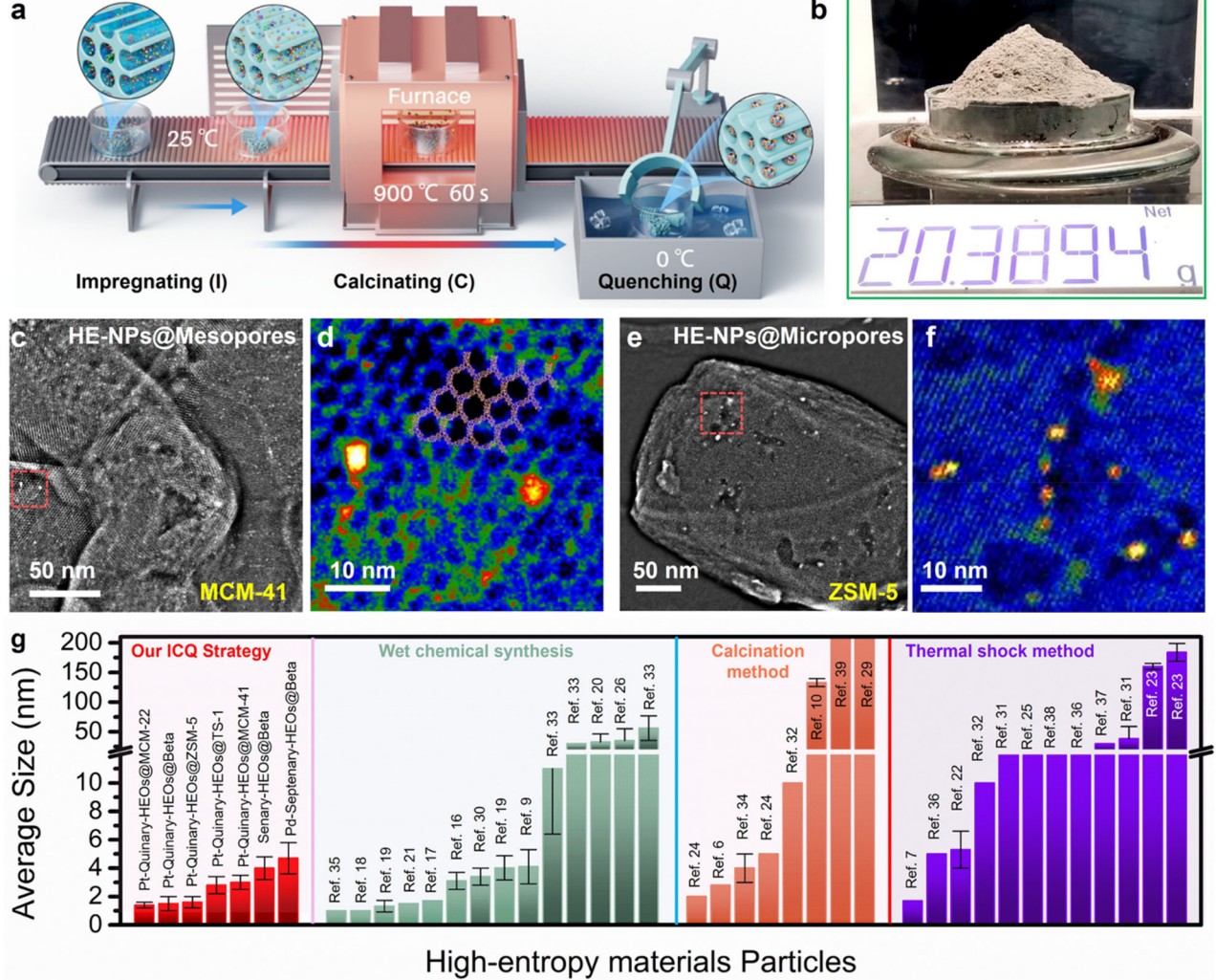

**Fig. 2 | HE-NPs@MSs: synthesis protocol and key features. a** Schematic diagram of the ICQ synthesis process, including incipient wetness impregnating (I), calcinating in seconds (C), and quenching (Q). **b** Photograph of as-prepared Senary-HEOs@MCM-41 materials. 20 g of $M_3O_4$@MCM-41 (containing Mn, Fe, Co, Ni, Cu, Zn) are prepared via the ICQ process within 5 min. Large-area HAADF-STEM image and HR HAADF-STEM image with color maps of **c, d** Pt-Quinary-HEOs@MCM-41, **e, f** Pt-Senary-HEOs@ZSM-5. **g** Comparison of nanoparticle size and uniformity of high entropy nanomaterials obtained by our ICQ strategy and other synthesis methods in previous works (see Supplementary Table 3 for details).

Supplementary Fig. 18a, when the precursor is calcinated and slowly cooled (Anneal), non-uniform multi-phase metal species form on the surface of MCM-41, with some nanoparticles exceeding 50 nm in size. Even after a quenching process for rapid cooling, the non-uniformity of the produced nanoparticles remains evident (Fig. 3b and Supplementary Fig. 18b). In contrast, the incipient wetness impregnation route ensures that all metal salt precursors are inhaled into the nanopores driven by the capillary effect. This process leads to the in-situ formation of liquid metal droplets in the nanopores during the high-temperature calcination, promoting more uniform nanoparticle formation. When the system is cooled slowly, the obtained material still contains multi-phase metal nanoparticles, affording uneven size distributions (Fig. 3c and Supplementary Fig. 18c). In the ICQ process, the liquid metal droplets are "frozen" without undergoing phase separation, resulting in the formation of HE-NPs within the nanopores. The confinement effect of the nanopores ensures that the size of the HE-NPs remains remarkably homogeneous (Fig. 3d). The crystal phase of as-prepared HE-NPs@MSs (i.e., Pt-Quinary-HEOs@MCM-41) via ICQ strategy is probed by the powder X-ray diffraction (PXRD) measurements. As depicted in Fig. 3e, the broad peak between 15 and 25° is contributed by the MCM-41 support, while the quite weak peaks could

be ascribed to the cubic-phase $M_3O_4$. It is noted that the low signal-to-noise ratio caused by the ultrafine confined HE nanoparticles, in combination with the strong signal from the MCM-41 support, could greatly reduce the accuracy of Scherrer equation-based estimation in HE particle sizes. The synchrotron-based small-angle XRD pattern demonstrates that the ordered nanopore structure could be maintained (Fig. 3e inset). Atomically-resolved STEM image in Fig. 3f shows an interplanar spacing of 0.25 and 0.24 nm with an angle of 100°, corresponding to the (311) and (222) planes of an fcc structure (Supplementary Fig. 19). Figure 3g manifests the atomic intensity distribution of the atomic columns (I and II) marked by rectangular regions in Fig. 3f. The disordered intensity fluctuations are suggestive of random distributions of metal atoms along crystal facets, further implying the formation of HE-NPs.

To gain insights into the thermodynamic and kinetic essence of ICQ strategy, we systematically investigated the effect of key parameters upon the morphology and chemistry of produced HE-NPs. Vapor pressure is of significance to the thermodynamic state and kinetic diffusion rate. To examine the pressure effect, we performed the ICQ reaction in a vacuum condition by sealing the precursor in a quartz tube (Fig. 4a). As shown in Supplementary Fig. 20, XRD pattern

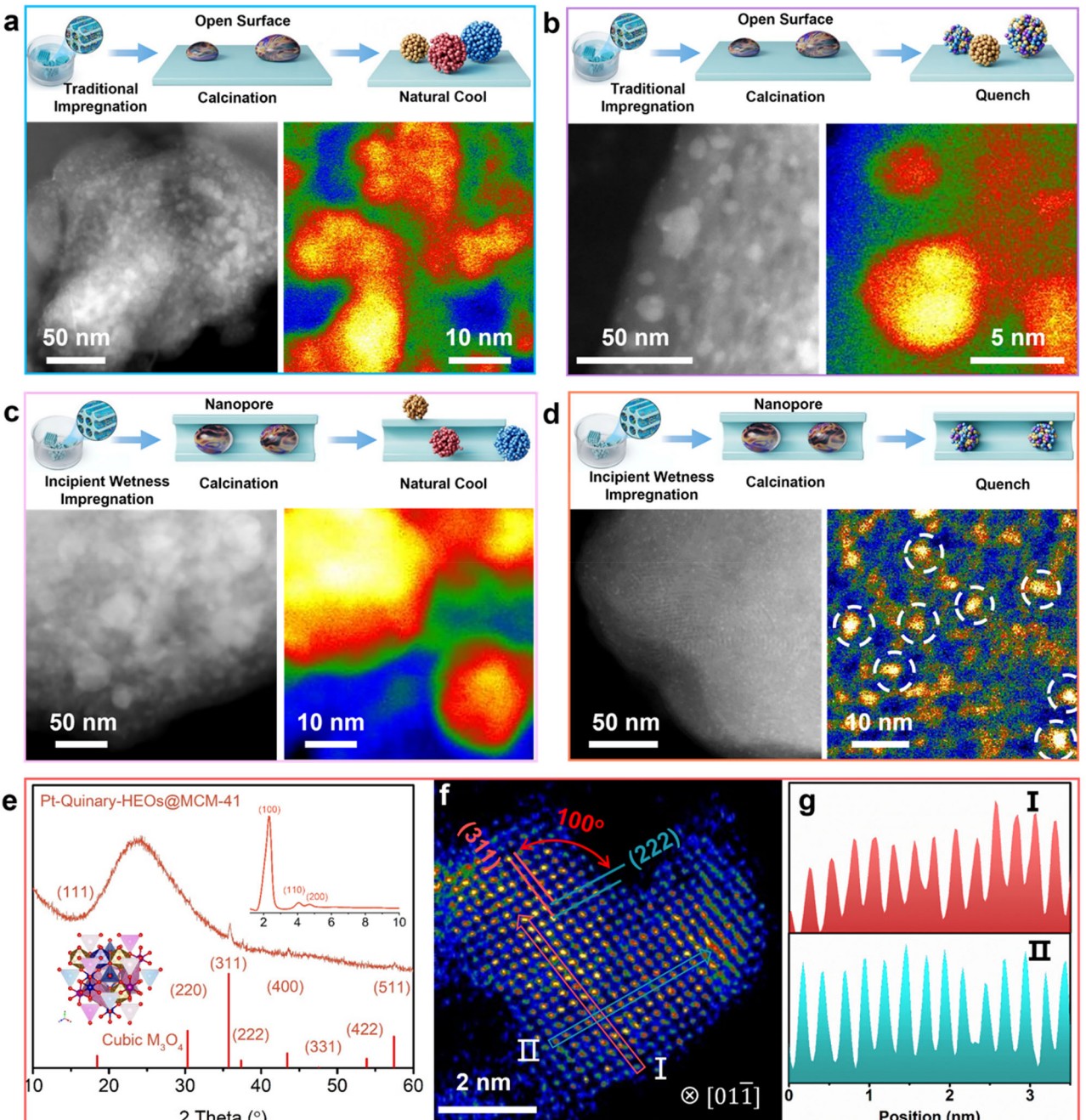

**Fig. 3 | Synthesis control. a–d** Schematic diagram of the synthesis of multiphase oxides (MPOs)/MSs and HEOs@MSs samples, with corresponding large-area/HR HAADF-STEM images and color maps. Pt-Quinary-MPOs/MCM-41 **a** synthesized by traditional impregnation, calcination, and annealing process; **b** synthesized by traditional impregnation, calcination, and quenching process; **c** synthesized by incipient wetness impregnation, calcination, and annealing process; **d** Pt-Quinary-HEOs@MCM-41 synthesized by ICQ route. **e–g** Structure of Pt-Quinary-HEOs@MCM-41: **e** XRD pattern (Insets: small-angle XRD pattern and cubic $M_3O_4$ model). **f** Atomic resolution STEM image of Pt-Quinary-HEOs nanoparticle. **g** Corresponding atomic intensity line profiles of atomic rows in (**f**).

indicates that the resultant octonary high-entropy alloy (HEA) sample harvests a single-phase cubic structure[22]. STEM observation and EDS mapping in Fig. 4b show that eight metal elements are uniformly dispersed within the nanoparticles, consistent with the XRD analysis. In contrast, the particle size exceeding 50 nm is markedly larger than that of its counterpart synthesized under normal pressure, which could be attributed to the higher vapor pressure of liquid metal under vacuum conditions, rendering the liquid droplets toward expansion.

We next investigated the effect of MS substrates. Cations typically reside within the pores to balance the negative charge. However, these cations can partially obstruct the nanopores, thereby influencing the accessibility and spatial distribution of other species, such as liquid metal droplets, and consequently affecting the formation of HE-NPs within the pores (Fig. 4c). For instance, when NaY is used as the growth support, where $Na^+$ cations reside inside the zeolite Y crystals, the resulting material exhibits particulate aggregates exceeding 50 nm in size. These aggregates are predominantly located on the outer surface of the NaY, rather than within the pores (Fig. 4d and Supplementary Fig. 21). Subsequently, NaY zeolite was converted to $NH_4Y$ zeolite through ion exchange, replacing $Na^+$ with $NH_4^+$ cations. After calcination, only small $H^+$ cations remain inside the Zeolite Y crystals (donated as HY). When HY is used as the support, the same ICQ route,

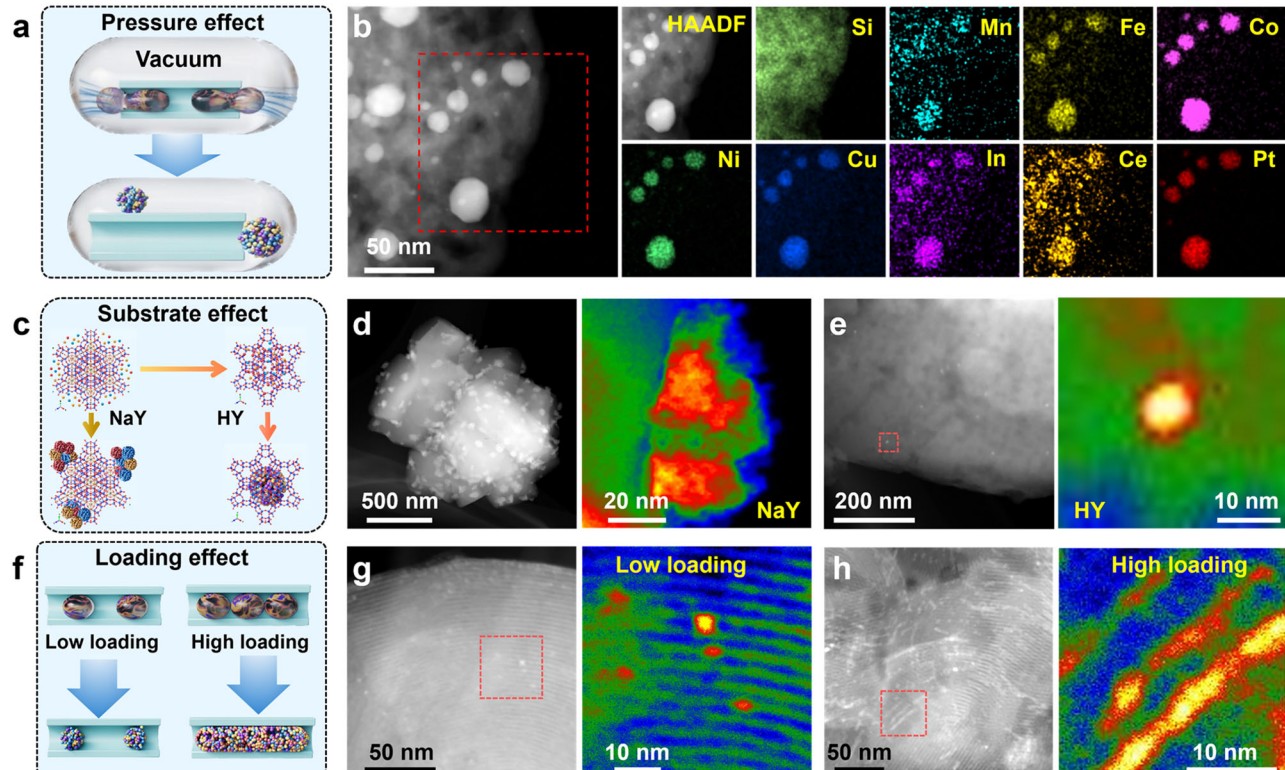

**Fig. 4 | Study on thermodynamic and kinetic conditions during the ICQ synthesis. a, b** Schematic diagram of sealing the precursor inside a vacuum quartz tube, and STEM images of obtained Pt-Septenary-HEAs@MCM-41. The influence of Na+ blocking the pores (NaY) and Na+ being replaced by H+ (HY): Schematic diagram (**c**), Large-area HAADF-STEM image and detailed HAADF-STEM image with color maps of (**d**) Pt-Quinary-MPOs/NaY, and (**e**) Pt-Quinary-HEOs@HY. The generation of HEOs-nanorods@MS by increasing metal loading: Schematic diagram (**f**), large-area HAADF-STEM image, and detailed HAADF-STEM image with color maps of Pt-Septenary-HEOs@MCM-41 materials with (**g**) 1.7 wt% metal (dispersed HEO nanoparticles) and (**h**) 3.5 wt% metal (HEO nanorods).

as expected, results in the formation of HE-NPs with a homogeneous size distribution within the nanopores, thanks to the reduced obstruction caused by the smaller H+ cations, as revealed in Fig. 4e and Supplementary Figs. 22–24.

Additionally, we further investigated the effect of precursor loading on the geometric characteristics of the HE-NPs. The FSLB simulation in Supplementary Fig. 25 and Supplementary Movie 6 suggest that in non-wetting nanopores, liquid metal droplets can evolve into 1D nano-columns as the precursor loading increases (Fig. 4f). At a low loading, the metal species grows into uniform HE-NPs with an average size of 3.0 nm in the pores via ICQ strategy (Fig. 4g and Supplementary Fig. 26). As the loading increases, the aggregated metal droplets give rise to the formation of rod-shaped HE-NPs along the one-dimensional nanopores (Fig. 4h and Supplementary Fig. 27). This phenomenon might offer an opportunity for preparing HE nanowires with controllable components and sizes.

## Generality of ICQ strategy

The ICQ strategy introduced here comprises a generic and versatile method for the preparation of HE-NPs@MSs. This is also applicable to high-throughput synthesis, which is crucial for developing HE-NPs@MSs materials libraries. Utilizing homemade high-throughput synthesis equipment (Supplementary Fig. 28), we managed to prepare various typed HE-NPs@MSs products in one batch. XRD analysis and HAADF-STEM observation demonstrate the fabrication of binary, quinary, octonary, denary, or undenary HE-NPs within MCM-41 (Supplementary Figs. 29–37). These materials encompass spectra of dissimilar elements, s-block (Mg, Sr), d-block (Mn, Fe, Co, Ni, Cu, Zn), and f-block (Ce), which provide abundant adsorption sites and diverse electronic structures for potential catalytic process.

To further verify the universality of this strategy, we extended the synthesis of HE-NPs across a variety of molecular sieve supports, affording micropores. As shown in Fig. 5a and Supplementary Fig. 38, Pt-Quinary-HE-NPs with an average particle size of $1.6 \pm 0.4$ nm are confined within the micropores of ZSM-5 zeolites. Figure 5b display Pt-Quinary-HE-NPs with an average particle size of $2.5 \pm 0.5$ nm within TS-1 zeolites (Supplementary Figs. 39 and 40), which has the same MFI topological structure with ZSM-5, yet with Ti atomic substitution in frameworks. Using MCM-22 zeolites with an MWW topology and 2D sheet morphology as supports, Pt-Quinary-HE-NPs exhibit an average particle size of $1.4 \pm 0.2$ nm (Fig. 5c and Supplementary Figs. 41 and 42). Similarly, highly dispersed Pt-Quinary-HE-NPs ($1.5 \pm 0.5$ nm) and Pd-Septenary-HE-NPs ($4.7 \pm 1.1$ nm) can also be confined to grow within Beta zeolites, which feature an intergrowth topology, as shown in Fig. 5d and Supplementary Figs. 43–49. Therefore, the ultrafine HE-NPs are located within the micropore-based MSs, in which the framework could effectively suppress the migration of metal species and enhance sinter-resistance ability under specific reaction conditions.

## High-temperature catalytic feature of HE-NPs@MSs

In high-entropy materials, multiple metal elements randomly occupy the same lattice regimes, leading to the emergence of unique physicochemical properties because of the complex elemental interactions[9,22,29]. The oxygen species in HEOs are expected to lose their distinct identity as part of a mono-elemental oxide, and instead exhibit relatively averaged properties. As shown in Supplementary Figs. 50–53 and Supplementary Table 4, the oxygen vacancy formation energies ($E_{Vo}$) demonstrate that the derived $E_{Vo}$ values of single metal oxides and bimetallic oxides vary greatly, from −6.7 eV for NiO to 21.6 eV for $Fe_2O_3$, while the $E_{Vo}$ with different chemical environments in

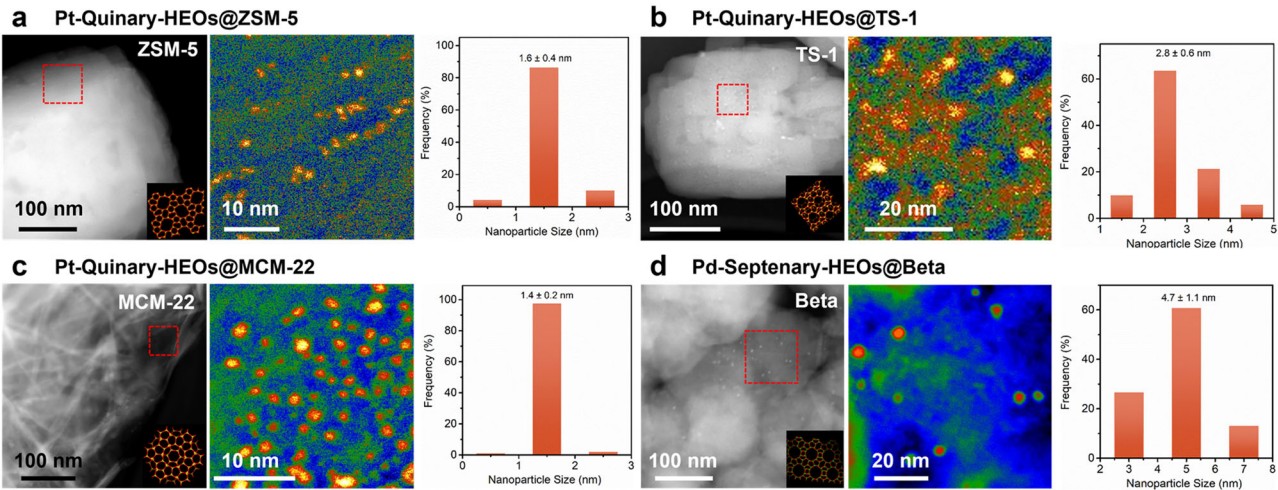

**Fig. 5 | Structure characterization of HE-NPs@MSs with a variety of molecular sieves.** Large-area HAADF-STEM image (Inset: zeolite model) and detailed HAADF-STEM image with color maps of **a** Pt-Quinary-HEOs@ZSM-5, **b** Pt-Quinary- HEOs@TS-1, **c** Pt-Quinary-HEOs@MCM-22, **d** Pd-Septenary-HEOs@Beta, as well as corresponding size distributions of HEO nanoparticles.

Senary-HEOs changes slightly between 3.4 and 4.1 eV. The $H_2$-temperature-programmed reduction ($H_2$-TPR) of as-prepared HEOs@MSs shows a main peak at 450 °C, but Senary-MPOs/MCM-41 exhibits multiple reduction peaks from 200 to 800 °C (Supplementary Figs. 54 and 55), which is consistent with the DFT calculation results. This altered environment affords potential to impact catalytic performances and physicochemical properties of the material[40].

Benefiting from the high-entropy effects and the confinement effect of the MSs matrix, as-prepared HE-NPs@MSs catalysts are expected to reduce the content of precious metals, possess anti-sintering properties, and exhibit favorable high activity and selectivity under harsh catalytic reaction conditions. As a proof of concept, we demonstrated Pt-Quinary-HEOs@MCM-41 as an advanced catalyst for PDH, a promising approach to meet the growing global demand for propylene and hydrogen[8,14,15,41,42]. The catalytic performances of different catalysts for PDH conversion were investigated at 550 °C with a high weight hourly space velocity of 13.5 $h^{-1}$ without co-feeding $H_2$. As shown in Supplementary Figs. 56 and 57, an initial propane conversion of 9.5% with a 98.7% propylene selectivity is achieved over Pt-Quinary-HEOs@MCM-41. In comparison, the Pt-Quinary-MPOs/MCM-41 catalyst shows a propane conversion of 2.8% with an 88.9% selectivity, while the Pt@MCM-41 catalyst exhibits only 0.5% propane conversion under the same condition (WHSV = 13.5 $h^{-1}$, 550 °C). The Quinary-HEOs@MCM-41 catalyst without Pt and Pt@MCM-41 catalyst with Pt nanoparticle dosage manifest no activity (Supplementary Figs. 58–60), indicative of the synergistic effect of Pt sites and HEOs in propane conversion. The formation rate (FR) of propylene over Pt-Quinary-HEOs@MCM-41 reaches up to 44.2 $mol_{propylene}$ $g_{Pt}^{-1}$ $h^{-1}$, which is 4.7 and 31.6 times higher than those of Pt-Quinary-MPOs/MCM-41 (9.5 $mol_{propylene}$ $g_{Pt}^{-1}$ $h^{-1}$) and Pt@MCM-41 (1.4 $mol_{propylene}$ $g_{Pt}^{-1}$ $h^{-1}$), respectively (Fig. 6a). This performance represents a top-tier level among all state-of-the-art Pt-based catalysts reported to date (Supplementary Table 5)[14,43–48]. Notably, after 365 min on stream, the propylene formation rate of Pt-Quinary-HEOs@MCM-41 remains constantly high at 30.9 $mol_{propylene}$ $g_{Pt}^{-1}$ $h^{-1}$ with a 97.8% propylene selectivity (Supplementary Fig. 56). In contrast, the formation rate of Pt-Quinary-MPOs/MCM-41 dropped sharply to 1.5 $mol_{propylene}$ $g_{Pt}^{-1}$ $h^{-1}$ with a 61.9% propylene selectivity under the identical conditions.

To elucidate the role of each metal in the high-entropy system, a series of catalysts were prepared by sequentially removing one element from the (Pt-MnFeCoCuIn)$O_x$@MCM-41 composition, with their performance evaluated in PDH reaction (Supplementary Figs. 61 and 62). The results demonstrate that Fe species suppress initial activity but enhance long-term stability; Mn, Co, or Cu species are essential for maintaining catalytic activity; and In species play the pivotal role in preserving both activity and selectivity. Collectively, these results underscore the importance of synergistic interactions among multiple metal elements in achieving a balanced combination of activity, stability, and selectivity—an inherent advantage of the high-entropy strategy.

Figure 6b and Supplementary Movie 7 show the 3D reconstruction of the Pt-Quinary-HEOs@MCM-41 (spent) catalysts, which demonstrate that the metal nanoparticles remain confined within the molecular sieve crystal upon PDH catalysis. The XZ- and YZ-slice in Fig. 6c, d clearly show that the metal nanoparticles (red spots) are embedded inside the MCM-41 (blue section). The elemental mappings and line scan profiles depict the homogeneous distribution of multiple metal species within individual nanoparticles of Pt-Quinary-HEOs@MCM-41 (spent) (Supplementary Figs. 63 and 64), with no significant change in particle size, indicating excellent sintering resistance. The thermogravimetric and CO-DRIFTS result indicates that the observed deactivation is mainly due to carbon deposition, rather than sintering or phase separation (Supplementary Figs. 65 and 66). As for the Pt-Quinary-MPOs/MCM-41 (spent), the HAADF-STEM observation showcases a significant boost in the size of the Pt-multiphase oxides, growing from 30 to 300 nm upon 365 min on stream (Supplementary Fig. 67). EDS characterizations reveal that separated-phase metal species aggregates on the surface of MCM-41. Additionally, layered graphitized carbon could be observed over the particle surface, suggestive of coke formation during the high-temperature PDH reactions.

X-ray absorption spectroscopy analysis was carried out to elucidate the electronic structure and coordination environment of Pt-Quinary-HEOs@MCM-41 and Pt-Quinary-HEOs@MCM-41 (spent). The Pt $L_{III}$ edge X-ray absorption near edge structure (XANES) spectra in Fig. 6e demonstrate that the Pt species are in oxidized state in Pt-Quinary-HEOs@MCM-41, and the valence state of Pt in spent catalyst is mainly in metallic state. The Fourier-transformed extended XAFS (EXAFS) spectra, curve fitting (Fig. 6f; Supplementary Figs. 68 and 69 and Supplementary Table 6), and wavelet-transformed EXAFS plots (Fig. 6g, h and Supplementary Fig. 70) unveil the coordination environment of Pt. The results reveal the presence of Pt–O–M (where M is non-Pt metal) in Pt-Quinary-HEOs@MCM-41, indicating that Pt species are atomically dispersed. In terms of Pt-Quinary-HEOs@MCM-41 (spent), Pt–M coordination is

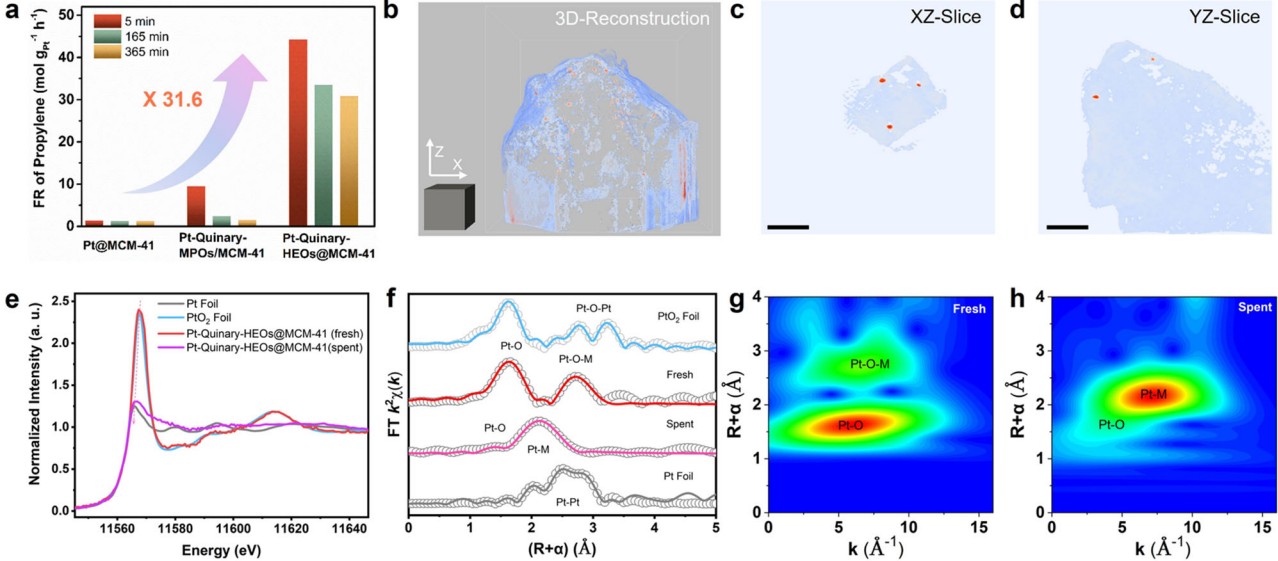

**Fig. 6 | Performances of HE-NPs@Zeolites in propane dehydrogenation. a** The formation rate of propylene over various catalysts. Reaction conditions: 0.1 g of catalyst mixed with 1.0 g of quartz sand, atmospheric pressure, $C_3H_8/N_2$ = 12.5/37.5 mL min$^{-1}$, WHSV = 13.5 h$^{-1}$, 550 °C. **b**–**d** Electronic tomography of Pt-Quinary-HEOs@MCM-41 (spent). **b** 3D-reconstruction (Scale cube, 50$^3$ nm$^3$), **c**, **d** XZ-slice

and YZ-slice (Scale bars, 50 nm). **e** Normalized XANES, **f** Fourier-transformed magnitude of EXAFS at the Pt $L_{III}$-edge, and **g**, **h** wavelet-transformed EXAFS plots of Pt-Quinary-HEOs@MCM-41 (fresh) and Pt-Quinary-HEOs@MCM-41 (spent). The $PtO_2$ and Pt foil were used as references.

observed, suggesting that the Pt species remain atomically dispersed without significant sintering after being subjected to the harsh PDH conditions. These collectively indicate that, in the Pt-Quinary-HEOs@MCM-41 catalyst, the MSs exert a confinement effect to stabilize the HE-NPs, while the non-noble metals in the HE-NPs help disperse the noble metal species, collaboratively catalyzing the PDH reaction.

## Discussion

The synthesis strategy via quenching space-confined liquid metal droplets offers a powerful approach to creating a broad library of anti-sintered ultrafine high-entropy nanoparticles within nanopores of molecular sieves. The general route involves an incipient wetness impregnation (I), a short-time (~60 s) calcination (C), and a rapid quenching (Q), which is beneficial to high-throughput screening of catalysts and could be integrated with large-scale preparation of industrial catalysts. These key features render ICQ a valuable platform for exploring novel materials and optimizing their exciting properties, bringing about a new repertoire of HE-NPs@MSs with unprecedented functionalities.

## Methods

### Typical synthesis

To prepare HEOs@MCM-41, 1 ml of mixed metal salt solutions is added dropwise to 1 g of MCM-41 and thoroughly mixed in the incipient wetness impregnation process. For the synthesis of HE-NPs@ molecular sieves (MSs), 1 ml of mixed metal salt solutions is added dropwise to 3 g of ZSM-5, NaY, HY, TS-1, MCM-22, Beta, and thoroughly mixed in the incipient wetness impregnation process. Then, the molecular sieves with mixed metal salt solutions adsorbed by capillary effect are dried and selectively retain the metal salt species in the pores. The metal-salts@MSs precursors are calcinated in a furnace preheated to 900 °C for about 60 s, and then quickly taken out and quenched in ice water.

### High-throughput synthesis

Fifty micrograms of metal-salts@MS precursors are placed in a self-made quartz tube group and calcined in a high-temperature furnace

preheated to 900 °C for about 60 s, and then quickly taken out and quenched in ice water.

### Material characterization

Wide-angle X-ray diffraction patterns were collected on a Bruker D8 with Cu Kα radiation (40 kV, 40 mA) at room temperature. The X-ray absorption data at the Pt $L_3$-edge of the samples were recorded at room temperature in the fluorescent mode with a silicon drift fluorescence detector at beam line BL14W1 of the Shanghai Synchrotron Radiation Facility (SSRF), China. The XAS was Small-angle X-ray diffraction patterns were performed at the 1W1A beam line of Beijing Synchrotron Radiation Facility (BSRF). Transmission electron microscopy (TEM), scanning transmission electron microscopy (STEM) images, electronic tomography, and corresponding energy dispersive X-ray spectroscopy (EDS) mapping were obtained with a Talos F200X instrument equipped with four EDS signal detectors. Atomic-resolution STEM images were carried out in an aberration-corrected Titan Themis G2 (FEI). The metal loading of the HE-NPs@MS and MPOs/MS was determined with Inductively Coupled Plasma-Optical Emission Spectrometer (ICP-OES) analyses carried out on an Agilent 5100 instrument. Nitrogen adsorption/desorption measurements were carried out on a Micromeritics 2020 analyzer at 77.35 K after the samples were degassed at 350 °C under vacuum. The temperature-programmed reduction of hydrogen ($H_2$-TPR) experiments were performed using a Micromeritics AutoChem II 2920 automated chemisorption analysis unit equipped with a thermal conductivity detector (TCD) under helium flow. Typically, 100 mg of the catalyst was pre-treated with He (30 mL/min) for 1 h at 300 °C, followed by cooling down to 50 °C; then the sample was reduced in a flow of 10% $H_2$/He (50 mL/min) mixture from 50 to 900 °C with a heating rate of 10 °C/min.

### Catalytic testing

The PDH experiments were performed in a quartz tubular fixed-bed reactor with a 13 mm inner diameter at atmospheric pressure. Before the dehydrogenation reaction, 0.1 g catalyst mixed with 1 g quartz sand was reduced at 550 °C under a $H_2$ flow of 50 mL for 30 min, and then fed with propane. Typically, the reaction gas mixture contained 25 vol

% propane and a balance of $N_2$ ($C_3H_8/N_2 = 12.5/37.5$ mL min$^{-1}$), which gave a WHSV of 13.5 h$^{-1}$. The reaction products were analyzed by an online gas chromatograph equipped with a flame ionization detector.

## Numerical methods

To describe the flow property and the interface behavior of liquid metal droplet, the governing equations, i.e., Navier−Stokes equations and Cahn−Hilliard equation, are given as (Eqs. 5–7):

$$\frac{\partial \rho}{\partial t} + \nabla(\rho \boldsymbol{u}) \tag{5}$$

$$\frac{\partial \boldsymbol{u}}{\partial t} + \nabla(\boldsymbol{uu}) = -\frac{1}{\rho}\nabla p + \nabla\left[\nu \cdot \left(\nabla \boldsymbol{u} + (\nabla \boldsymbol{u})^T\right)\right] + \frac{1}{\rho}\boldsymbol{F}_s, \tag{6}$$

$$\frac{\partial C}{\partial t} + \nabla(C\boldsymbol{u}) = M\nabla^2 \mu_C + Q_m, \tag{7}$$

where $\rho$, $\mathbf{u}$, $p$ and $\nu$ are the fluid density, velocity, pressure and kinematic viscosity, respectively. The surface force term $\boldsymbol{F}_s$ can be calculated by $\boldsymbol{F}_s = \mu_C \nabla C$, $C$ is the order parameter ranging from 0 to 1, and $\mu_C$ is the chemical potential. The mass correction term $Q_m$ imposed on the interface is used to maintain the mass conservation of the system. The details of the Surface energy calculation and Numerical Methods are in the Supplementary Information.

## Data availability

The data that support the findings of this study are available within the article and its Supplementary Information files.

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

## Acknowledgements

This work was supported by the National Natural Science Foundation of China (22179089, 22205153, 12302361, 12402328), China Postdoctoral Science Foundation (2024M762282), Jiangsu Funding Program for Excellent Postdoctoral Talent, Guangdong Basic and Applied Basic Research Foundation (2022A1515110174), and Shenzhen Science and Technology Innovation Commission (RCBS20221008093107026, RCBS20231211090722037). We thank D.H. (SUSTech) and L.X. (Great Bay University) for help in the TEM measurements, K.T. (Soochow University), and Y.Q. (JLU) for discussions on the behavior of metal droplets in confined spaces. The authors also gratefully acknowledge the cooperation of Y.C. and other beamline scientists at BSRF-1W1A beamline (https://cstr.cn/31109.02.BSRF. 1W1A). We thank the Shanghai Synchrotron Radiation Facility of BL14W1 beamline (https://cstr.cn/1124.02.SSRF.BL14W1) for the assistance on XAS measurements.

## Author contributions

S.C., X.L., and J.S. conceived the idea and performed the material synthesis. S.C. performed the TEM measurements. S.C. and Y.G. conducted the 3D reconstruction. J.S., Z.L., Q.S. and S.D. supervised the project. Z.Q. and Q.S. carried out the catalytic test. X.L., Z.-Q.D. and P.Y. performed the FSLB simulation. P.-F.L. performed the DFT calculations. S.C. and J.S. prepared the manuscript with input from all co-authors.

## Competing interests

The authors declare no competing interests.
