## [Peer Review file · Nature Communications]

Space-confined synthesis of sinter-resistant high-entropy nanoparticle library

Corresponding Author: Professor Jingyu Sun

Version 0:

Reviewer comments:

Reviewer #1

(Remarks to the Author)

The authors report the HE-NP synthesis in the confined space. Rapid synthesis, controlled size distribution, and high conversion of propane dehydrogenation are realized.

The reviewer found some of authors' claims lack the rational support.

Authors applied lattice boltzmann method to schematically illustrate the morphological change of nanoparticle in nanopore. LB method is based on continuum theory, thus its application is valid for micrometer scale particles, for example. In the case of nanoalloy, some specific elements with low surface energy may segregate on the surface, some specific elements with affinity to oxygen may segregate on the interface with matrix oxide. While presented simulation results provide a tendency in general, but does not support the sinter-resistance of synthesized HE-NPs.

Homogeneity of element distribution within NP is not clear. With overlays of EDX mappings of each constituent elements, one can clearly understand homogeneous high-entropy material are synthesized. In addition, this is effective to show the stability of high-entropy material after annealing or catalytic tests. The characterization of HE-NP is essentially important to discuss its stability and catalytic activity.

Because the catalytic properties degrade with time, there should be a certain change in catalyst such as sintering, phase separation of a certain elements, and surface deactivation by deposition of a certain species.

Some of figures are difficult to interpret due to the lack of appropriate labels. For example, color maps associated with STEM images do not make sense unless appropriately described.

Reviewer #2

(Remarks to the Author)

The submitted work describes a high-throughput impregnation-calcination-quenching (ICQ) strategy for synthesizing high-entropy nanoparticles (HE-NPs) confined in molecular sieves for constructing sinter-resistant, catalytically active systems. The concept of combining high-entropy compositions with space-confinement in nanoporous materials is original and addresses an important challenge in the development of thermally stable catalysts. The approach is broadly applicable, and experimentally well-supported. The authors provide comprehensive characterization of the resulting materials and demonstrate impressive performance in propane dehydrogenation (PDH), supported by both experimental and computational data. However, while the manuscript contains high-quality data and meaningful results, some scientific aspects need clarification or improvement to meet the publication standards of Nature Communications. The following issues should be addressed in a revision process.

(1) Although the authors emphasize the tunability of compositions (quinary to Septenary systems), it is unclear whether the catalytic performance is affected by the type or number of metals. The compositional effect on catalytic performance needs to be discussed more clearly.

(2) Although the PDH reactions with series of Pt-based catalysts are compelling, direct evidence supporting the sintering resistance of Pt sites is still limited. The authors provide post-reaction STEM and PXRD data showing that the particle morphology is largely retained. However, to support the claim of sintering resistance more convincingly, the authors are encouraged to perform CO-chemisorption measurements before and after PDH reaction, which can provide a quantitative assessment of metal dispersion and accessible surface area. In addition, X-ray photoelectron spectroscopy (XPS) on Pt 4f levels could help evaluate the chemical state and stability of Pt species under reaction conditions. Given that the structural stability of HE-NPs is a key highlight of the study, incorporating such analyses would significantly strengthen the manuscript.

(3) The manuscript repeatedly refers to the synthesized materials as high-entropy nanoparticles (HE-NPs). However, the claim of entropic stabilization and true atomic-level configurational disorder is not rigorously validated. EDS and PXRD confirm elemental co-location and single-phase nature, but do not confirm random substitution at atomic lattice sites. Atomic-resolution STEM, while informative, does not resolve the statistical distribution of elements on crystallographic positions. To substantiate the high-entropy nature, the authors would use extended X-ray absorption fine structure (EXAFS) analysis, atom probe tomography (APT) or synchrotron XAS.

(4) The authors demonstrate superior performance of Pt-containing HE-NPs@MS catalysts in PDH, but provide no mechanistic rationale for the observed enhancement. Claims of cocktail effect and entropy-modulated oxygen activation are vague and speculative without direct evidence. Clarification of the role of Pt versus other elements in the reaction mechanism is needed.

In overall, the authors successfully demonstrate a novel synthetic approach for generating a library of sinter-resistant, high-entropy nanoparticles with excellent catalytic performance. The concept of combining compositional entropy and space-confinement is both timely and relevant. However, to strengthen the impact of manuscript and ensure reproducibility and clarity, the points listed above must be adequately addressed. This reviewer recommends major revision before the manuscript is considered for publication in Nature Communications.

Reviewer #3

(Remarks to the Author)

This is an interesting manuscript that proposes potentially valuable material, particularly within the field of catalysis. However, the characterisation does not fully support the conclusion drawn. Specifically in relation to the nature of the high entropy nanoparticles or their precise location within the materials

The following points need to be irrefutable if the manuscript is to be publishable.

- While the use of EDX has confirmed the presence of the range of elements within a single support particle, it does not (nor can it at the resolution analysed) confirm the presence of all elements within a single nanoparticle. This is especially critical for those systems in which TEM indicates that the sizes are below the threshold of XRD detection. High resolution atomic level EDX imaging of individual sub 2-3 nm nanoparticles needs to be undertaken over a range of particles to confirm uniformity of composition.
- Where XRD peaks from metal oxide phases are detected, these should be analysed using the Scherrer equation to allow comparison to sizes from TEM. Moreover, it should be made clear if reflection from the metal oxide phase is observed in the zeolite systems, for which the majority illustrate solely reflections from the zeolite framework (or at least appear to do so).
- Characterisation of the Pt@MCM-41 is required to confirm that the nature of the species is indeed comparable with Pt-Quinary-HEOs@MCM-41. In particular, with regard to sizes and proposed location.
- The location of high-energy nanoparticles within the pores of the MCM-41 and zeolites requires electron tomography to categorically confirm they are located within the pores rather than on the surface of the support. I accept that sinter resistance is a strong indicator of confinement; however, given that it is not extensively supported by a range of characterisation techniques (e.g., surface vs bulk elemental analysis, detailed pore volume analysis), and relies solely on 2d projections from TEM, it is not categorical proof.

Version 1:

Reviewer comments:

Reviewer #1

(Remarks to the Author)

The reviewer would like to thank the authors' effort to respond to the reviewers' comments.

I would recommend the authors to explicitly describe the limitations of LBM.

The reviewer's point is not if it is for mesoscopic or macroscopic, but it is not discrete modeling like MD as the authors mention in their response.

LBM is a class of computational fluid dynamics methods for fluid simulation. Fluid is a continuum medium, and LBM is based on an approximation to particle distribution function for generated lattices. This method is valid when the focused entity is homogeneous, and its behavior is statistically the same.

Therefore, the reviewer described as below.

In the case of nanoalloy, some specific elements with low surface energy may segregate on the surface, some specific elements with affinity to oxygen may segregate on the interface with matrix oxide. While presented simulation results provide

a tendency in general but does not support the sinter-resistance of synthesized HE-NPs.

The reviewer would point out that the peer-review process is not personal communication between the reviewer and authors, thus it is typical that the authors reflect the reviewers' concern by revising the manuscript itself or make a full rebuttal. This time, the reviewer would encourage the authors to explicitly discuss the limitation of applying LBM to the nanoscale systems in the main body or supplementary of the manuscript.

Reviewer #2

(Remarks to the Author)

The revised manuscript looks sufficient for publication in this journal. The authors improved all the necessary issues reflecting all reviewers' comments, and therefore it is recommended for acceptance.

Reviewer #3

(Remarks to the Author)

The authors had addressed the majority of my concerns, and I now recommend that the manuscript be accepted for publication.

Response to Reviewers' Comments (NCOMMS-25-17756-T)

Reviewer #1

The authors report the HE-NP synthesis in the confined space. Rapid synthesis, controlled size distribution, and high conversion of propane dehydrogenation are realized.

The reviewer found some of authors' claims lack the rational support.

Our response: We are grateful for the reviewer's positive and constructive review. All the concerns based on the professional comments from the reviewer have been addressed in the revised manuscript.

1. Authors applied lattice boltzmann method to schematically illustrate the morphological change of nanoparticle in nanopore. LB method is based on continuum theory, thus its application is valid for micrometer scale particles, for example. In the case of nanoalloy, some specific elements with low surface energy may segregate on the surface, some specific elements with affinity to oxygen may segregate on the interface with matrix oxide. While presented simulation results provide a tendency in general, but does not support the sinter-resistance of synthesized HE-NPs.

Reply 1: We thank the reviewer for the insightful comment. The simulation only reveals a general tendency showing different behaviors of metal precursor droplets on open surfaces versus within confined pore spaces. This provides a rational explanation for the formation of uniform and ultrafine nanoparticles inside nanopores, while larger and less uniform particles tend to form under similar conditions outside the pores.

We also appreciate your professional comments on our numerical methodology. In response to your concerns, we have provided a comprehensive reply on the essence of the LB method in the present study.

i. Clarification on the Theoretical Nature and Applicable Scales of the LB Method

The LB method originates from the discretization of the Boltzmann transport equation.¹⁻³ Its core lies in describing fluid behavior through particle distribution functions, instead of relying on the continuum hypothesis. Therefore, it fundamentally belongs to a mesoscopic modeling approach within the statistical physics framework,

rather than a traditional continuum model, such as the Navier-Stokes equations. In other words, LB equation is based on the particle distribution function and statistical physics.

When the simulation scale is much larger than the molecular mean free path, the LB model can rigorously derive the Navier-Stokes equations through statistical averaging. Consequently, it is equivalent to continuum theory. The continuum equivalence of the LB method gradually loses validity at this scale, which is in line with your insightful judgment. To accurately calculate the behavior of nanoalloys, it requires direct characterization using microscopic models like Molecular Dynamics (MD). However, implementing MD simulations for this study would entail substantial computational resources and excessively long runtime, presenting practical limitations for the current scope of research.

ii. Positioning and Simulation Value of the LB Method in This Study

In light of the aforementioned calculation boundaries, the principal aim of employing the recently developed fractional step LB (FSLB) method in this research is to capture the morphological evolution trends of nanoparticles confined within nanopores, rather than delving into the elemental segregation mechanisms at the atomic scale. Compared with full-atom simulations, e.g., MD simulations of 100 nm particles require millions of core-hours, the LB method reduces computational costs by 2-3 orders of magnitude while maintaining key physical mechanisms. Besides, the FSLB method retains more characteristics derived from the particle distribution function and statistical physics. It simulates interface behaviors through the evolution of equilibrium and non-equilibrium distribution functions. By employing FSLB method, the thermodynamically driven spontaneous growth process is simulated, and the trends of final droplet size with diffusion kinetics, growth time, and precursor concentration are qualitatively investigated.

We would like to clarify that the simulation is primarily intended to illustrate the formation process of HE-NPs rather than to directly demonstrate their sinter-resistance. The observed sinter-resistance of the synthesized HE-NPs is mainly attributed to the anchoring and spatial confinement effects offered by the molecular sieve support.

Supporting References:

1. Qian, Y. H., D. D. Humières, P. Lallemand, Lattice BGK Models for Navier-Stokes Equation. *Europhys Lett* **17**, 479 (1992).
2. Shan, X., H. Chen, Lattice Boltzmann model for simulating flows with multiple phases and components. *Phys Rev E* **47**, 1815-1819 (1993).
3. Chen, S., G. D. Doolen, LATTICE BOLTZMANN METHOD FOR FLUID FLOWS. *Annual Review of Fluid Mechanics* **30**, 329-364 (1998).

2. Homogeneity of element distribution within NP is not clear. With overlays of EDX mappings of each constituent elements, one can clearly understand homogeneous high-entropy material are synthesized. In addition, this is effective to show the stability of high-entropy material after annealing or catalytic tests. The characterization of HE-NP is essentially important to discuss its stability and catalytic activity.

Reply 2: We appreciate the reviewer's valuable suggestion regarding the importance of elemental homogeneity within individual HE-NPs to support the stability of high-entropy materials. Indeed, we fully agree with the reviewer that such a characterization is crucial. It is worth noting that carrying out the EDX mapping on individual sub-3 nm HE-NPs is technically challenging, where the nanoparticles are embedded within the confined channels of molecular sieves, and the total metal loading is only ~2 wt%. In the work, by tuning the gas pressure during the synthesis, larger-sized HE-NPs could be obtained, for which individual-particle EDX mapping were performed and the elements are uniformly distributed within single particles.

In the revised manuscript, we have further supplemented nanoscale EDX analysis to strengthen this point. To minimize sample drift, we first stabilized the sample in the TEM for an extended period (2 hours) and then performed EDX acquisition over a prolonged duration (~30 minutes) using a Talos TEM equipped with four EDX detectors. The resulting mapping and line scan profiles of Pt-Quinary-HEOs@MCM-41 (**Fig. S8**) demonstrate that multiple metal elements are homogeneously distributed within individual particles. To elucidate the structural stability of the catalyst, EDX analysis was conducted on the samples after the catalytic reaction. The elemental mappings and line scan profiles depicted the homogeneous distribution of multiple metal species within individual nanoparticles of Pt-Quinary-HEOs@MCM-41-spent (**Fig. S64**). These results demonstrate that the nanoparticles in Pt-Quinary-HEOs@MCM-41 and Pt-quinary-HEOs@MCM-41 (spent) are high-entropy nanoparticles.

We have updated these results in the revised manuscript as **Fig. S8** and **Fig. S64** with related interpretations. We copy the revision here for your kind check:

Fig. S8 Elemental mapping and corresponding line scan profiles for nanoparticles of Pt-Quinary-HEOs@MCM-41 (fresh).

Fig. S64 Elemental mapping and corresponding line scan profiles for nanoparticles of Pt-Quinary-HEOs@MCM-41-spent.

3. *Because the catalytic properties degrade with time, there should be a certain change in catalyst such as sintering, phase separation of a certain elements, and surface deactivation by deposition of a certain species.*

Reply 3: We thank the reviewer for the constructive comment with respect to any possible structural/compositional changes during catalysis. To address this concern, we have supplemented detailed characterizations of the fresh-catalyst and spent-catalyst.

Electron tomography confirms that the HE-NPs remain confined within the mesoporous channels of the molecular sieve upon PDH catalysis, with no significant change in particle size, indicating excellent sintering resistance (**Fig. 5b–d; Movie S7**). EDS analysis did not reveal any noticeable phase separation among the constituent elements (**Fig. S64**). The CO diffuse reflectance infrared Fourier transform spectroscopy (CO-DRIFTS) results confirm that Pt exists in a single-atom state and is exposed on the surface for the case of both fresh- and spent-catalyst (**Fig. S66**). The overall decrease of CO-DRIFTS in adsorption intensity was attributed to the color change induced by coke. Furthermore, XAS analysis suggests that Pt remains atomically dispersed within the Pt-Quinary-HEOs@MCM-41-spent (**Fig. 5e–h**).

Coke formation is one of the main factors leading to the decline in catalytic performance during PDH reaction. The coke analysis of Pt-Quinary-HEOs@MCM-41-spent demonstrates that the total weight loss is *ca.* 30% (**Fig. S65**), and the carbon deposits can only be completely removed above 600 °C. The desorption below 300 °C is attributed to dehydration, while that above 300 °C is due to coke combustion. A coke deposition rate of 16.28 mg_{coke} g_{cat}⁻¹ h⁻¹ was measured, indicating that the catalyst is prone to coking during the reaction. This suggests that the observed deactivation is mainly due to carbon deposition, rather than sintering or phase separation.

We have updated these results in the revised manuscript as **Fig. 5**, **Fig. S64**, **Fig. S65**, and **Fig. S66** with related interpretations. We copy the revision here for your kind check:

Fig. 5b–d The electron tomography of Pt-Quinary-HEOs@MCM-41-spent. The color map of segmented reconstruction and representative tomograms. Scale cube, 50³ nm³; Scale bar, 50 nm.

Fig. S64 Elemental mapping and corresponding line scan profiles for nanoparticles of Pt-Quinary-HEOs@MCM-41-spent.

Fig. S65 Thermogravimetric analysis of Pt-Quinary-HEOs@MCM-41-spent.

Fig. S66 CO-DRIFTS spectra of Pt-Quinary-HEOs@MCM-41 and Pt-Quinary-HEOs@MCM-41-spent.

4. *Some of figures are difficult to interpret due to the lack of appropriate labels. For example, color maps associated with STEM images do not make sense unless appropriately described.*

Reply 4: We thank the reviewer for the detailed advice. Following the reviewer's suggestion and referring to relevant literature (*Nat. Catal.* 2020, 3, 628), we have revised the figure annotations to explicitly indicate the type of STEM images. Specifically, we have labeled them as "large-area HAADF-STEM images" and "HR HAADF-STEM images with color maps" to improve clarity and facilitate interpretation. We believe these modifications will make the figures more informative and easier to be understood (for example, **Fig. 4**).

Reviewer #2

The submitted work describes a high-throughput impregnation-calcination-quenching (ICQ) strategy for synthesizing high-entropy nanoparticles (HE-NPs) confined in molecular sieves for constructing sinter-resistant, catalytically active systems. The concept of combining high-entropy compositions with space-confinement in nanoporous materials is original and addresses an important challenge in the development of thermally stable catalysts. The approach is broadly applicable, and experimentally well-supported. The authors provide comprehensive characterization of the resulting materials and demonstrate impressive performance in propane dehydrogenation (PDH), supported by both experimental and computational data. However, while the manuscript contains high-quality data and meaningful results, some scientific aspects need clarification or improvement to meet the publication standards of Nature Communications. The following issues should be addressed in a revision process.

In overall, the authors successfully demonstrate a novel synthetic approach for generating a library of sinter-resistant, high-entropy nanoparticles with excellent catalytic performance. The concept of combining compositional entropy and space-confinement is both timely and relevant. However, to strengthen the impact of manuscript and ensure reproducibility and clarity, the points listed above must be adequately addressed. This reviewer recommends major revision before the manuscript is considered for publication in Nature Communications.

Our response: We are grateful for the reviewer's positive and constructive review. All the concerns based on the professional comments from the reviewer have been addressed in the revised manuscript.

- 1. Although the authors emphasize the tunability of compositions (quinary to Septenary systems), it is unclear whether the catalytic performance is affected by the type or number of metals. The compositional effect on catalytic performance needs to be discussed more clearly.*

Reply 1: Thanks for your insightful comment. We have incorporated clearer discussion on how the type and number of metal components affect the catalytic performance in the revised manuscript. Our response is summarized below:

1) Effect of multimetallic composition on Pt dispersion and stabilization: It was reported that multicomponent environments would promote the dispersion of Pt species and stabilize their atomically dispersed state (*J. Am. Chem. Soc.* 2022, 144, 15944). In

the revised manuscript, we have included additional characterizations of the Pt@MCM-41 sample, showing that Pt exists as nanoparticles with sizes at ~4 and 8 nm (**Fig. S58**). In the high-entropy Pt-containing system (Pt-MnFeCoCuIn@MCM-41), the XAS results indicate that Pt exists in an atomically dispersed form (**Fig. 5f**). These findings suggest that the high-entropy composition inhibits Pt aggregation and promotes atomic dispersion, which is beneficial to the catalytic activity.

Fig. S58 HAADF-STEM image of Pt@MCM-41 and element map of Pt.

Fig. 5f Fourier-transformed magnitude of EXAFS at the Pt L_{III} -edge of Pt-Quinary-HEOs@MCM-41 (fresh) and Pt-Quinary-HEOs@MCM-41 (spent). The PtO₂ and Pt foil were used as references.

2) Effect of individual metal components on PDH performance: To investigate the role of each metal in the high-entropy system, we prepared a series of catalysts by removing one element at a time from the (Pt-MnFeCoCuIn)_x@MCM-41 composition. The corresponding catalysts, (Pt-FeCoCuIn)_x@MCM-41, (Pt-

MnCoCuIn) O_x @MCM-41, (Pt-MnFeCuIn) O_x @MCM-41, (Pt-MnFeCoIn) O_x @MCM-41 and (Pt-MnFeCoCu) O_x @MCM-41, were tested for propane dehydrogenation. The results are presented in Fig. S61 and Fig. S62. Key observations include:

Removal of Fe increased the initial conversion rate to 19.8%, but the conversion declined to 4.3% after 365 min, indicating that Fe suppresses initial activity but improves stability.

Removal of Mn, Co, or Cu led to a decrease in initial conversion (5.0–7.3%) and further decline to 1.5–2.5% after 365 min, suggesting that these elements contribute positively to catalytic activity.

Removal of In resulted in an initial conversion of 6.6%, which dropped to 0.6% after 365 min. Moreover, the selectivity decreased significantly from ~90% to ~68.5%, indicating that In plays a crucial role in maintaining both activity and selectivity.

These results collectively demonstrate the critical role of compositional tuning in optimizing the catalytic performance. The synergistic effects of multiple metal elements enable a balance between activity, stability, and selectivity, which is a key advantage of the high-entropy approach. The relevant discussions and experimental data have been incorporated into the revised manuscript accordingly.

Fig. S61 Effect of individual metal components of (Pt-MnFeCoCuIn) O_x @MCM-41 on PDH performance: Conversion of propane over various catalysts by removing one element at a time from the (Pt-MnFeCoCuIn) O_x @MCM-41. Reaction conditions: 0.1 g of catalyst mixed with 1.0 g of quartz sand, atmospheric pressure, $C_3H_8/N_2 = 12.5/37.5$ mL min^{-1} , $WHSV = 13.5$ h^{-1} , 550 °C.

Fig. S62 Effect of individual metal components of (Pt-MnFeCoCuIn)_x@MCM-41 on PDH performance: Propylene selectivity over various catalysts by removing one element at a time from the (Pt-MnFeCoCuIn)_x@MCM-41. Reaction conditions: 0.1 g of catalyst mixed with 1.0 g of quartz sand, atmospheric pressure, C₃H₈/N₂ = 12.5/37.5 mL min⁻¹, WHSV = 13.5 h⁻¹, 550 °C.

2. *Although the PDH reactions with series of Pt-based catalysts are compelling, direct evidence supporting the sintering resistance of Pt sites is still limited. The authors provide post-reaction STEM and PXRD data showing that the particle morphology is largely retained. However, to support the claim of sintering resistance more convincingly, the authors are encouraged to perform CO-chemisorption measurements before and after PDH reaction, which can provide a quantitative assessment of metal dispersion and accessible surface area. In addition, X-ray photoelectron spectroscopy (XPS) on Pt 4f levels could help evaluate the chemical state and stability of Pt species under reaction conditions. Given that the structural stability of HE-NPs is a key highlight of the study, incorporating such analyses would significantly strengthen the manuscript.*

Reply 2: We sincerely appreciate your constructive suggestion, which has allowed us to strengthen the manuscript. Following these comments, detailed structural analyses of the samples before and after the PDH reaction were supplemented.

The electron tomography confirms that the HE-NPs remain well confined within the mesoporous channels of the molecular sieve, with no significant changes in particle size after the reaction, indicating good sintering resistance (**Fig. 5b-d**; **Movie S7**). The

elemental mappings and line scan profiles in **Fig. S64** depict the homogeneous distribution of multiple metal species within individual nanoparticles of Pt-Quinary-HEOs@MCM-41-spent, and no significant element segregation is observed. Furthermore, CO diffuse reflectance infrared Fourier transform spectroscopy (CO-DRIFTS) in **Fig. S66** demonstrates that Pt remains in a single-atom state and is exposed on the surface both before and after the reaction, supporting the high dispersion and accessibility of Pt sites.

Because of the confinement of Pt nanoparticles within the mesoporous channels with a relatively low Pt content, it is challenging to accurately detect the chemical state of Pt species using XPS as advised. X-ray absorption spectroscopy (XAS) was in target employed to investigate the chemical state and stability of Pt species confined within the nanopores (**Fig. 5f**; **Fig. S68**; **Table S6**). For the fresh catalyst, the first coordination shell of Pt consists of O atoms, and the second shell corresponds to a Pt–O–M (M is non-Pt metal). For the spent catalyst, a small amount of Pt–O coordination is observed, while Pt–M coordination is predominant. The coordination number of Pt-M is 5.1, which is significantly lower than the Pt–Pt coordination number of 12 in Pt foil. The bond length of Pt-M (2.56 Å) is also shorter than that of Pt–Pt (2.76 Å).

The supplemental data and discussions have been incorporated into the revision, clearly demonstrating the structural stability of HE-NPs and substantially strengthening our manuscript.

Fig. 5f Fourier-transformed magnitude of EXAFS at the Pt L_{III} -edge of Pt-Quinary-HEOs@MCM-41 (fresh) and Pt-Quinary-HEOs@MCM-41 (spent). The PtO₂ and Pt foil were used as references.

Fig. 5b-d The electron tomography of Pt-Quinary-HEOs@MCM-41-spent. The color map of segmented reconstruction and representative tomograms. Scale cube, 50^3 nm^3 ; Scale bar, 50 nm.

Fig. S64 Elemental mapping and corresponding line scan profiles for nanoparticles of Pt-Quinary-HEOs@MCM-41-spent.

Fig. S66 CO-DRIFTS spectra of Pt-Quinary-HEOs@MCM-41 and Pt-Quinary-HEOs@MCM-41-spent.

3. *The manuscript repeatedly refers to the synthesized materials as high-entropy nanoparticles (HE-NPs). However, the claim of entropic stabilization and true atomic-level configurational disorder is not rigorously validated. EDS and PXRD confirm elemental co-location and single-phase nature, but do not confirm random substitution at atomic lattice sites. Atomic-resolution STEM, while informative, does not resolve the statistical distribution of elements on crystallographic positions. To substantiate the high-entropy nature, the authors would use extended X-ray absorption fine structure (EXAFS) analysis, atom probe tomography (APT) or synchrotron XAS.*

Reply 3: We sincerely thank you for this insightful comment. As you are pointing out, atomic-level configurational disorder is an intrinsic characteristic of high-entropy materials and underpins their unique physicochemical/catalytic properties. We fully agree with the reviewer that advanced techniques such as atom probe tomography (APT), synchrotron-based X-ray absorption spectroscopy (XAS), and extended X-ray absorption fine structure (EXAFS) analysis are essential for validating the statistical distribution of elements and the truly random substitution at crystallographic sites in high-entropy systems (*Nature* 2023, 624, 564). These characterization methods are not only critical for confirming the high-entropy nature but also vital for understanding the structure–property relationships. Nonetheless, each of these techniques entails substantial experimental/analytical complexity and often requires dedicated, standalone investigations. Therefore, it is difficult to comprehensively incorporate all of them within the scope of the present work.

Following the reviewer’s suggestion, we have carried out synchrotron-based XAS measurements to investigate the local coordination environment of Pt species in a representative high-entropy catalyst (**Fig. S68; Table S6**). XANES and EXAFS analyses reveal that: i) In the fresh catalyst, the first coordination shell of Pt consists of O atoms, with a coordination number (CN) of 5.4, which is lower than that in PtO₂ (CN=6); The second shell corresponds to a Pt–O–M (M is non-Pt metal) structure, with a coordination number (CN) of 10.1, which differs from the Pt–O–Pt coordination number (CN) of 6.1 in PtO₂. The results indicate atomic-level dispersion of Pt within the high-entropy oxide lattice. ii) After the PDH reaction, XAS results show predominant Pt–M bonding in the first shell, with a coordination number of 5.1, significantly lower than the Pt–Pt coordination number of 12 in Pt foil. The Pt–M bond length (2.56 Å) is also shorter than those in metallic Pt (2.76 Å). These results suggest the presence of defects and structural disorder around Pt sites, further supporting the presence of stable, active Pt single-site species within the high-entropy nanoparticles.

These findings strongly support that Pt is atomically dispersed and stabilized within the high-entropy nanoparticles and further highlight the critical role of local disorder in governing catalytic activity. The corresponding data and discussion have been included in the revised manuscript. Once again, we thank the reviewer for the valuable suggestion. In future work, we will expand our efforts to incorporate more detailed EXAFS modeling and atom-probe tomography to systematically probe the formation mechanisms of atomic-level configurational disorder and its correlation with the properties of high-entropy materials.

Fig. 68 R-space of EXAFS and the fitting plots of Pt-Quinary-HEOs@MCM-41(fresh) and Pt-Quinary-HEOs@MCM-41 (spent). The PtO₂ and Pt foil were used as references.

4. The authors demonstrate superior performance of Pt-containing HE-NPs@MS catalysts in PDH, but provide no mechanistic rationale for the observed enhancement. Claims of cocktail effect and entropy-modulated oxygen activation are vague and speculative without direct evidence. Clarification of the role of Pt versus other elements in the reaction mechanism is needed.

Reply 4: Thank you for pointing this out. We agree that understanding the mechanistic contributions of Pt and other constituent metals is critical. In the revised manuscript, we have supplemented the experimental data and mechanistic discussion as follows:

1) Confirmation of atomically dispersed Pt as the active center: Propane dehydrogenation tests show that the sample of (MnFeCoCuIn) O_x @MCM-41 without Pt exhibits a conversion rate below 1%, indicating that Pt is essential for catalytic activity in the (Pt-MnFeCoCuIn) O_x @MCM-41. Further structural characterization of Pt@MCM-41 using STEM and CO-DRIFTS confirmed that Pt exists primarily as nanoparticles (4–8 nm), as shown in **Fig. S58 and S59**. Despite the presence of Pt, the Pt@MCM-41 sample showed a propane conversion rate below 1%, suggesting that large Pt particles alone are much less active under these conditions. In the (Pt-MnFeCoCuIn) O_x @MCM-41 catalyst, CO-DRIFTS and XAS results reveal that Pt is atomically dispersed due to the high-entropy effect. These findings confirm that atomically dispersed Pt serves as the active site, while other metals dilute and modulate Pt species electronically and structurally, contributing synergistically to the PDH reaction by cocktail effect.

Fig. S58 HAADF-STEM image of Pt@MCM-41 and element map of Pt.

Fig. S59 and S66 CO-DRIFTS spectra of Pt-Quinary-HEOs@MCM-41 and Pt@MCM-41.

2) Clarifying the role of each element in the reaction mechanism: To further clarify the distinct roles of Pt versus other elements, we conducted additional PDH tests on a series of catalysts by removing one element at a time from the (Pt-MnFeCoCuIn) O_x @MCM-41 composition. The corresponding catalysts— (Pt-FeCoCuIn) O_x @MCM-41, (Pt-MnCoCuIn) O_x @MCM-41, (Pt-MnFeCuIn) O_x @MCM-41, (Pt-MnFeCoIn) O_x @MCM-41 and (Pt-MnFeCoCu) O_x @MCM-41—were tested for propane dehydrogenation (PDH). The results are presented in **Fig. S61 and S62**. Key observations include:

Removal of Fe increased the initial conversion rate to 19.8%, but the conversion declined to 4.3% after 365 min, indicating that Fe suppresses initial activity but improves stability.

Removal of Mn, Co, or Cu led to a decrease in initial conversion (5.0–7.3%) and further decline to 1.5–2.5% after 365 min, suggesting that these elements contribute positively to catalytic activity.

Removal of In resulted in an initial conversion of 6.6%, which dropped to 0.6% after 365 min. Moreover, the selectivity decreased significantly from ~90% to ~68.5%, indicating that In plays a crucial role in maintaining both activity and selectivity.

These results collectively demonstrate the critical role of compositional tuning in optimizing the catalytic performance. The synergistic effects of multiple metal elements enable a balance between activity, stability, and selectivity, which is a key advantage of the high-entropy approach. The catalytic mechanism of high-entropy nanoparticles for PDH is indeed complex and scientifically significant, and previous studies have made valuable exploratory contributions in this area (*J. Am. Chem. Soc.* 2022, 144, 15944; *Angew. Chem. Int. Ed.* 2025, 64, e202419093). We sincerely appreciate the reviewer's suggestion. In our future work, we will conduct in-depth investigations into the catalytic mechanism to establish clear structure–activity relationships.

Reviewer #3

This is an interesting manuscript that proposes potentially valuable material, particularly within the field of catalysis. However, the characterisation does not fully support the conclusion drawn. Specifically in relation to the nature of the high entropy nanoparticles or their precise location within the materials.

The following points need to be irrefutable if the manuscript is to be publishable.

Our response: We are grateful for the reviewer's positive and constructive review. All the concerns based on the professional comments have been addressed in the revised manuscript.

- 1. While the use of EDX has confirmed the presence of the range of elements within a single support particle, it does not (nor can it at the resolution analysed) confirm the presence of all elements within a single nanoparticle. This is especially critical for those systems in which TEM indicates that the sizes are below the threshold of XRD detection. High resolution atomic level EDX imaging of individual sub 2-3 nm nanoparticles need to be undertaken over a range of particles to confirm uniformity of composition.*

Reply 1: We greatly appreciate your valuable suggestion regarding the importance of confirming elemental homogeneity within individual HE-NPs. We fully agree that such high-resolution characterization is essential to support the formation and stability of high-entropy materials. It is worth noting that performing EDX mapping on individual sub-3 nm HE-NPs is technically challenging, in which the nanoparticles are embedded within the confined channels of molecular sieves, and the total metal loading is only around 2 wt%.

In the revised manuscript, we have supplemented nanoscale EDX analysis to further reinforce this point. To reduce sample drift, the specimen was stabilized in the TEM for an extended period (~2 h), followed by prolonged EDX acquisition (~30 min) using a Talos TEM equipped with four EDX detectors. As shown in **Fig. S8**, the resulting EDX mapping and line scan profiles confirm that multiple metal elements are co-located within single nanoparticles, supporting their identity as high-entropy materials. This result has been included in the revised manuscript.

Fig. S8 Elemental mapping and corresponding line scan profiles for nanoparticles of Pt-Quinary-HEOs@MCM-41 (fresh).

2. *Where XRD peaks from metal oxide phases are detected, these should be analysed using the Scherrer equation to allow comparison to sizes from TEM. Moreover, it should be made clear if reflection from the metal oxide phase is observed in the zeolite systems, for which the majority illustrate solely reflections from the zeolite framework (or at least appear to do so).*

Reply 2: Thank you for this insightful comment. It is noted that the low signal-to-noise ratio caused by the ultrafine confined HEO nanoparticles, in combination with the strong signal from the MS support, could greatly influence the peak broadening and reduce the accuracy of Scherrer equation-based estimation toward the particle sizes. In this regard, our extensive STEM characterization across multiple regions did not reveal any large-sized particles, where the atomically-resolved STEM image shows an interplanar spacing of 0.25 and 0.24 nm with an angle of 100° , corresponding to the (311) and (222) planes of an fcc structure (for example, in **Fig. 2f**). In addition, we have supplemented nanoscale EDX analysis to further reinforce this point (**Fig. S8**).

As for the zeolite systems, we have supplemented the XRD patterns of the pristine zeolite and HEOs@Zeolites for comparison. As shown in **Fig. S49**, the diffraction peaks of the HEOs are not clearly visible in the HEOs@Zeolite composites. This is

attributed to the low metal loading (~1.5 wt%) and ultrafine size of the HEO nanoparticles, whose weak diffraction signals are overwhelmed by the strong and complex reflections from the zeolite framework. Furthermore, the XRD patterns of the HEOs@zeolites closely resemble those of the pure zeolite, indicating that the overall structure of the zeolite framework is well preserved after incorporating HEOs. This observation is consistent with our TEM results and nitrogen adsorption/desorption analyses, which also confirm the structural integrity of the zeolite host. This result has been included in the revised manuscript.

Fig. S49 The PXR D pattern of (a) Pt-Quinary-HEOs@ZSM-5, (b) Pt-Quinary-HEOs@TS-1, (c) Pt-Quinary-HEOs@MCM-22, (d) Pd-Septenary-HEOs@Beta samples prepared by incipient wetness impregnation, calcination, and quench process (ICQ strategy).

3. Characterisation of the Pt@MCM-41 is required to confirm that the nature of the species is indeed comparable with Pt-Quinary-HEOs@MCM-41. In particular, with regard to sizes and proposed location.

Reply 3: Thank you for your constructive comment. In the revised manuscript, we have incorporated detailed characterization of the Pt@MCM-41 sample. As shown in **Fig. S58**, STEM images reveal that Pt exists as nanoparticles with a size distribution of 3–4

nm, which is consistent with the pore size of MCM-41. These particles are rarely found at the edges of the MCM-41 crystals on the 2d projections from TEM, suggesting that they are predominantly located within the mesopores. This observation is in line with the tomography results obtained for HEOs@zeolites, which show a comparable spatial distribution of nanoparticles within the molecular sieves matrix. A few larger particles (~8 nm) are observed on the outer surface, likely due to migration of Pt species out of the pores during thermal treatment, as the particle size exceeds the channel dimensions and no structural damage to the surrounding framework is observed. In addition, CO-DRIFTS analysis (**Fig. S59**) indicates the existence of Pt nanoparticles. As a control, we also prepared a Pt/MCM-41 sample via conventional impregnation (**Fig. S60**). In this case, most Pt nanoparticles are located on the external surface of the MCM-41 with sizes ranging from 10 nm to 100 nm.

The presence of Pt nanoparticles observed in Pt@MCM-41 indicates that, in the high-entropy oxide system, multiple metal species facilitate the dispersion and stabilization of Pt. Our related discussion has been added to the revised manuscript.

Fig. S58 HAADF-STEM images of Pt@MCM-41 and element map of Pt.

Fig. S59 CO-DRIFTS spectra of Pt@MCM-41.

Fig. S60 HAADF-STEM image of Pt/MCM-41 and element map of Pt.

4. *The location of high-energy nanoparticles within the pores of the MCM-41 and zeolites requires electron tomography to categorically confirm they are located within the pores rather than on the surface of the support. I accept that sinter resistance is a strong indicator of confinement; however, given that it is not extensively supported by a range of characterisation techniques (e.g., surface vs bulk elemental analysis, detailed pore volume analysis), and relies solely on 2d projections from TEM, it is not categorical proof.*

Reply 4: We sincerely thank you for this valuable suggestion. In the revised manuscript, we have supplemented electron tomography analysis for HEOs@MSs to directly confirm the confinement of high-entropy nanoparticles within the porous structures. The corresponding 3D reconstructions of Pt-Quinary-HEOs@MCM-41 and Senary-HEOs@ZSM-5 (**Fig. S5; Movie S4; Fig. S12; Movie S5**) clearly show that the metal nanoparticles are embedded inside the molecular sieve crystal particles rather than located on the external surface. The 3D reconstruction of the Pt-Quinary-HEOs@MCM-41-spent catalysts also demonstrate that the metal nanoparticles remain confined within the molecular sieve crystal after PDH catalysis, as shown in **Fig. 5** and **Movie S7**.

We hope that these supplementary evidences would strengthen our claim and enhance the overall quality of the manuscript. We appreciate your helpful comment.

Fig. S5 The electron tomography of Pt-Quinary-HEOs@MCM-41. The color map of segmented reconstruction and representative tomograms (The red dots represent metal particles). Scale cube, 50^3 nm^3 ; Scale bar, 50 nm.

Fig. S12 The electron tomography of Senary-HEOs@ZSM-5. The color map of segmented reconstruction and representative tomograms (The red dots represent metal particles). Scale cube, 50^3 nm^3 ; Scale bar, 50 nm.

Fig. 5b-d The electron tomography of Pt-Quinary-HEOs@MCM-41-spent. The color map of segmented reconstruction and representative tomograms. Scale cube, 50^3 nm^3 ; Scale bar, 50 nm.

Response to Reviewers' Comments (NCOMMS-25-17756A)

Reviewer #1 (Remarks to the Author):

The reviewer would like to thank the authors' effort to respond to the reviewers' comments.

I would recommend the authors to explicitly describe the limitations of LBM.

The reviewer's point is not if it is for mesoscopic or macroscopic, but it is not discrete modeling like MD as the authors mention in their response.

LBM is a class of computational fluid dynamics methods for fluid simulation. Fluid is a continuum medium, and LBM is based on an approximation to particle distribution function for generated lattices. This method is valid when the focused entity is homogeneous, and its behavior is statistically the same.

Therefore, the reviewer described as below.

In the case of nanoalloy, some specific elements with low surface energy may segregate on the surface, some specific elements with affinity to oxygen may segregate on the interface with matrix oxide. While presented simulation results provide a tendency in general but does not support the sinter-resistance of synthesized HE-NPs.

The reviewer would point out that the peer-review process is not personal communication between the reviewer and authors, thus it is typical that the authors reflect the reviewers' concern by revising the manuscript itself or make a full rebuttal. This time, the reviewer would encourage the authors to explicitly discuss the limitation of applying LBM to the nanoscale systems in the main body or supplementary of the manuscript. The authors report the HE-NP synthesis in the confined space. Rapid synthesis, controlled size distribution, and high conversion of propane dehydrogenation are realized.

Our response: We are grateful for the reviewer's positive and constructive review. All the concerns based on the professional comments from the reviewer have been addressed in the revised manuscript.

We sincerely appreciate the reviewer's thoughtful comments and the opportunity to clarify the limitations of our simulation methodology. We acknowledge that while the fractional step lattice Boltzmann (FSLB) method offers certain adaptability, it also has inherent limitations. In the revised manuscript, we have explicitly discussed the applicability and limitations of FSLB in both the main text and the Supplementary

Information. Our use of the FSLB method is intended to capture the qualitative morphological evolution of droplets on open surfaces and within nanopores. However, we recognize that FSLB is limited in accurately describing heterogeneous systems and fluid behavior at the nanoscale. Therefore, our simulations are qualitative. To quantitatively assess the behavior of nanoalloys—where atomic-level segregation, surface energy differences, or element-specific interactions (e.g., oxygen affinity) may significantly affect physical behavior—atomistic modeling approaches such as Molecular Dynamics (MD) are required.

In our revised manuscript, we have now included a clear statement highlighting this limitation:

It is worth noting that the FSLB approach used here primarily captures qualitative trends in morphological evolution on open surfaces and within nanochannels. For quantitative insights into the behavior of heterogeneous nanoalloys, such as the affinity between specific elements and the substrate, complementary approaches such as Molecular Dynamics (MD) simulations would be necessary.

This clarification has been added to the main text (page 5-6) and also briefly noted in the Supplementary Information (page 6). We thank the reviewer again for emphasizing this important point and helping us improve the clarity and scientific rigor of our manuscript.